# Development of Nanosensors Based Intelligent Packaging Systems: Food Quality and Medicine

**DOI:** 10.3390/nano11061515

**Published:** 2021-06-08

**Authors:** Ramachandran Chelliah, Shuai Wei, Eric Banan-Mwine Daliri, Momna Rubab, Fazle Elahi, Su-Jung Yeon, Kyoung hee Jo, Pianpian Yan, Shucheng Liu, Deog Hwan Oh

**Affiliations:** 1Department of Food Science and Biotechnology, College of Agriculture and Life Science, Kangwon National University, Chuncheon 24341, Korea; ericdaliri@yahoo.com (E.B.-M.D.); elahidr@gmail.com (F.E.); sujung0811@gmail.com (S.-J.Y.); ttt1528@naver.com (K.h.J.); pianpianyan1@gmail.com (P.Y.); 2College of Food Science and Technology, Guangdong Ocean University, Guangdong Provincial Key Laboratory of Aquatic Products Processing and Safety, Guangdong Province Engineering Laboratory for Marine Biological Products, Guangdong Provincial Engineering Technology Research Center of Marine Food, Key Laboratory of Advanced Processing of Aquatic Product of Guangdong Higher Education Institution, Zhanjiang 524088, China; weishuaiws@126.com; 3Collaborative Innovation Center of Seafood Deep Processing, Dalian Polytechnic University, Dalian 116034, China; 4School of Food and Agricultural Sciences, University of Management and Technology, Lahore 54770, Pakistan; rubab.momna@gmail.com

**Keywords:** drug management, iMedBox, in-home healthcare station (IHHS), wireless sensor network (WSN), controlled delamination material (CDM), radio frequency identification (RFID), intelligent packaging

## Abstract

The issue of medication noncompliance has resulted in major risks to public safety and financial loss. The new omnipresent medicine enabled by the Internet of things offers fascinating new possibilities. Additionally, an in-home healthcare station (IHHS), it is necessary to meet the rapidly increasing need for routine nursing and on-site diagnosis and prognosis. This article proposes a universal and preventive strategy to drug management based on intelligent and interactive packaging (I2Pack) and IMedBox. The controlled delamination material (CDM) seals and regulates wireless technologies in novel medicine packaging. As such, wearable biomedical sensors may capture a variety of crucial parameters via wireless communication. On-site treatment and prediction of these critical factors are made possible by high-performance architecture. The user interface is also highlighted to make surgery easier for the elderly, disabled, and patients. Land testing incorporates and validates an approach for prototyping I2Pack and iMedBox. Additionally, sustainability, increased product safety, and quality standards are crucial throughout the life sciences. To achieve these standards, intelligent packaging is also used in the food and pharmaceutical industries. These technologies will continuously monitor the quality of a product and communicate with the user. Data carriers, indications, and sensors are the three most important groups. They are not widely used at the moment, although their potential is well understood. Intelligent packaging should be used in these sectors and the functionality of the systems and the values presented in this analysis.

## 1. Introduction

To create better materials and products, nanostructure utilizes nanoscale design and device integration of existing materials. Nanomaterial applications have made their presence felt in a variety of fields, including healthcare, grafts, prostheses, smart fabrics, power generation and maintenance with power-producing constituents, and highly well-organized arrays, such as security, surveillance, sabotage, and shadowing [1]. The bio-nanomaterial study has developed as an incredible new field, known as a transdisciplinary borderline in materials science and the medical field. Nano-biochip materials, nanoscale biomimetic materials, nano-motors, nano-composite materials, and interface bio-materials are important in industrial, defense, and clinical medicine applications, and nano-biosensors and nano-drug-delivery systems have promising futures. Bio-molecular instruments, for example, are used in genetic engineering and diagnosis and have a large pharmaceutical potential for biosensor growth. Biomolecules play a vital part in nanoscience. The biosensor consists of a substrate for bio-sensing and a transducer was applied to identify biochemical agents. The target analytes are recognized explicitly by bio-sensing materials such as proteins, antibodies, DNA/RNA probes, while devices can quantitatively track the biochemical processes outlined in Figure 1.

Singh et al. [2] have documented a disposable biosensor for rapid determination of not only H_2_O_2_ but also azide using a biotic materials (polyaniline (PANI)) based bio-electrode. This film does an excellent job of maintaining enzyme activity and preventing leakage. It suggests that, as shown in Figure 2, this powerful film can also be used to immobilize, indicating a potential forum for biosensor growth [3].

Biosensors have emerged as a modern area of interdisciplinary study. Biosensors of different forms, each with its own set of strengths and drawbacks, are combined with various transmitters to form bio-sensing in the field of nanobiotechnology (lab-on-a-chip, nano-biosensor array) [4]. Furthermore, the nano-biosensor can be easily put into a single-use polymer-based chip for various biological analysis and medical diagnostics products. The perceptions of synchronized nano-biosensors that incorporate the desired possessions of the distinct constituents: susceptibility and linking accuracy polymer machinery, the chemistry of peptides or nucleic acid to align the different electron transducing units, and nano-electronics are recent developments in nano-biosensors and their application in ecology, especially in healthcare diagnosis. The results of these systems focus on the potential benefits of combining nanoscale diligence with biosensors to revolutionize medical diagnosis and management. Thus, the creation and application of nano-devices in biology and medicine will have enormous implications for humanity and healthcare. In agronomic production, nourishment processing, specialized treatment, and the environment, bio-sensors combined with evolving expertise in molecular nano-biology, microfluidics, and nanomaterials are used for the rapid, specific, sensitive, accessible, in-field, online, and real-time diagnosis and prevention of chemicals, medicines, microbes, toxins, proteins, bacteria, plants, livestock, and foods. Ultrasensitive bio-sensors provide an excellent analytical tool for monitoring systems, allowing the biosphere to be monitored. In the context of bio-electronics, nano-based molecular technology, and upcoming new advances in biosensor growth, they appear to be all upward fields that will have a significant impact on the development of a novel bio-sensing stage to solve our future medical diagnoses and report tasks related to pollution problems to healthcare, as well as all living circumstances. There is currently a lot of interest in deciphering genetic progressions at the solitary fragment stage and applying them to future nanobiotechnology applications. Studies of single molecule-enabled probes have opened up exciting research avenues, especially nanoscience, to improve versatile biomolecule-based [5] technology.

Nanomaterials have allowed the development of ultra-sensitive biosensors due to variations in their exterior to capacity ratio, surface characteristics, and ionic conductivity, as well as excellent stability. They were used to facilitate the biochemical reaction and convey a gesture of bio-recognition proceedings by targeting the transmission and wiring of metabolites to the electrode’s surface. Nanomaterials for bio-sensing have been published in many research papers, including nanoparticles, nanowire, nano-needle, nano-sheet, nanotube, nano-rod, and nano-belt [6], as shown in Figure 3. Modified carbon nanotube (CNT) electrodes, gold nanoparticles for ionic and immune sensors, nanostructured sensors [7,8], biosensing based on nanoparticles [9], and nanowire as sensing materials [10] are just a few significant examples.

## 2. Nanotechnology-Based Sensors for Food Analysis and Monitoring of Food Security

Better health is important for human happiness and well-being, as well as economic development [11]. As a result, the world’s attention is drawn to food safety concerns [12]. Many well-known food detection technologies are already used, such as chromatography, immunoassaysation, and catalytic hydrolysis [13,14,15]. However, these methods require lengthy and time-consuming detection processes, which limits their use in foods. A quick test is now available [16,17]. Nanosensors in the food processing chain allow for data curation, which ensures food safety [18]. The field of nanosomeschanism has research proposals on food so categorized because of its simplicity in both systems and ease of implementation.

### 2.1. Nutrients (Antioxidants and Sugars) Quantification 

Since vitamins and antioxidant components in foods degrade quickly during processing and storage, nanosensors have been used in the food industry to detect them. Folic acid (FA) is an essential part of the haemapoietic system, and its lack causes leucopenia, gigantocytic anemia, behavioral devolution and psychosis, heart attack, and carcinogenesis, etc. Rao et al. [19]. The identification of FA in fruit juices, wheat flour, and milk samples has been identified using multi-walled carbon nanotube (MWNT) and single-walled carbon nanotube (SWCNT)–ionic liquid nano-composites [20,21]. They are sensing essential amino acids, tryptophan, and vitamin ascorbic acid [22]. Likewise, NiO nanoparticles towards sensing of ascorbic acid [23]. Taei et al. [24] developed a magnetic nanoparticle-modified MWNT for instantaneous grit on ascorbic acid used in pharmaceutical and biological samples. The antioxidant potential of red wine can be due to its phenolic material. The tyrosinase enzyme was produced and used to determine the consistency of phenols (gallic acid, catechol, chlorogenic acid, caffeic acid, and protocatechualdehyde) in white wines using immobilized gold-based nanoparticles [25,26]. Nanosensors are also useful for assessing the maturity and ripening of fruit by measuring glucose, sucrose, and ascorbic acid concentrations [27]. 

### 2.2. Toxins Detection Andtoxin Producing Pathogen Detection

Microorganism produced toxins, which causes major health problems all over the world. Microorganisms produce toxins as an act of self-defense [28]. As a result, developing sensitive and fast methods to detect toxins in foods and related products is critical [29,30]. A carcinogenic toxin called aflatoxins is produced by food products infected with Aspergillus parasiticus and Aspergillus flavus. Gold nanoparticles decorated with anti-aflatoxin antibodies were used to detect these toxins in food.

Similarly, Agilent magnetic beads containing antiaflatoxin M1 antibodies and gold nano-probes were used to detect aflatoxin M1 in the milk sample [31,32,33,34]. Palytoxin, a form of marine toxin, is commonly found in seafood. Carbon-based nanotubes-luminous bio-sensors have been used to detect toxins in *Pteriomorpha* (marine mussels) meat [35].

### 2.3. Adulterants

Adulteration of food has a negative impact on human health and has become a global health problem. Food adulterants can cause anything from diarrhea and vomiting to paralysis and even death [28]. A competitive work is food adulterant at the lower stage, implemented via a repetitive recognition method. Melamine can be composted. However, as a contaminant examples include protein-enriched foods, beverages, food, and candies. Gold nanoparticles were created to locate melamine (in picomoles) [36]. Sudan I (red dye) was contaminated with red pepper powder, known to be carcinogenic in food [37].

### 2.4. Residual Veterinary Antibiotics and Pesticides

Chloramphenicol was first isolated from Streptomyces venezuelae and is a cost-effective but toxic broad-spectrum antibiotic used to treat infection in bees. Honey becomes infected as a result of the continued use of this antibiotic [28]. Chloramphenicol detection in honey samples was achieved using nanoparticles made from polyethylene glycol (PEG). For the selective detection of chloramphenicol in milk, Xiao [21] developed a hybrid molecularly imprinted polymer with a surface-enhanced Raman spectroscopy (SERS) sensor. Pesticides can already be detected in foods using nanoparticle-based nanosensors. Organophosphates are the most widely used pesticides [38]. Carbamate and organophosphorus pesticides have been detected using gold nanoparticles and fluorometric and colorimetric sensors [39]. Acetamiprid is a pesticide that poses a potential health risk to humans. Jokar et al. [40] developed a biosensor based on aptamer and silver nanoparticles to detect Acetamiprid. Ligand-free carbon quantum dots sensors (CQDs) can also detect pesticide residues in agricultural products [41].

### 2.5. Pathogens

The presence of pathogenic bacteria in foods is the leading cause of food spoilage and foodborne diseases. Identifying a whole bacterial cell or bacterial deoxyribonucleic acid (DNA) may be used to detect these bacteria [42]. The use of nanoparticles to help isolate DNA and detect bacteria has yielded promising results, although it was more sensitive and time-consuming than other methods [28]. Banerjee et al. [43,44] created multiparametric magneto-fluorescent nanosensors that could detect *Escherichia coli* (*E. coli*) O157:H7 contamination in less than an hour. The pathogenic Listeria monocytogenes bacterium was isolated from milk samples using magnetic iron oxide nanoparticles, and the DNA was then quantified using polymerase chain reaction (PCR) [45]. Varshney et al. [46] created magnetic nanoparticle-antibody conjugates and found that the magnetic nanoparticles captured *E. coli* O157:H7 more effectively in ground beef samples. In PCR-based microbial detection, the 16s ribosomal ribonucleic acid (rRNA) is primarily used as a selective marker. The 16s rRNA microarray process, on the other hand, is costly and insensitive, while the nanoparticle-based detection method is simple and accurate [47,48]. 

### 2.6. Heavy Metals

Heavy metal ions (Pb^2+^, Cd^2+^, Hg^2+^, Ag^+^, and Cu^2+^) in food and water have posed a danger to human safety and the climate. CQDs, organic fluorescent dyes, quantum dots, and rare earth metal ions have been combined to create ratiometric sensors for detecting Cu(II) in vegetable and fruit samples [49]. Babar et al. [50] developed a low-cost gold nano-textured electrode for detecting arsenic in food and water. For selective and responsive examination, this electrochemical gold nano-textured electrode can be used to detect arsenic in a complex system comprising Fe^2+^, Pb^2+^, Hg^2+^, Cu^2+^, Ni^2+^, and other ions [51]. Optical dual-mode nanosensors based on gold nanoparticles and CQDs could detect arsenic (ii) in water in a recent sample [52]. Based on the fluorescence resonance energy transfer between long-strand ap-tamers-functionalized upconversion nanoparticles and short-strand aptamers-functionalized gold nanoparticles, a turn-on nanosensor was designed to detect Hg^2+^ in tap water and milk samples and showed good selectivity and precision [52].

## 3. Application of Nanomaterials Inactive and Functional Packaging for Smart Packaging Concepts

Mainly with the advancement of nanotechnology, realistic and creative food packaging, such as intelligent packaging, is increasingly taking shape in response to the growing demand for food quality, protection, and shelf-life from customers and the food industry [53], which could extend and implement all the principles of packaging and provide novel material to help improve mechanical, barrier and antimicrobial properties as well as monitoring the food during transport and storage [54]. Different types of sensors, such as chemical sensors or biosensors, may be used in innovative packaging [55] to track the quality and protection of packaged goods like food, pharmaceuticals, or health and household products by measuring freshness, microbials, leakages, environmental gas composition, pH, time, or temperature [56,57], which is divided into active packaging and intelligent packaging [58]. 

### 3.1. Time–Temperatures Indicators 

Intelligent packaging tracks product quality and environmental conditions [53]. Time-temperature indicators (TTI) are simple, cost-effective, and simple to use for realtime monitoring of the impact of temperature on quality and safety from farm user [59,60]. TTI, also known as external markers because they are attached outside the box, tracked and analyzed a variety of food categories, including fruits [61], meat [62], milk [63], and marine products [64] under chilling or freezing conditions.

### 3.2. Antimicrobial Active Packaging

Antimicrobials included amino acids, bacteriocins, polysaccharides, and fungi and enzymes in the products [65]. Incorporating antimicrobials and an active packaging system allows for innovative packaging materials to be used to increase shelf life, consistency, and protection. Furthermore, as compared to normal counterparts, nano-sized antimicrobials demonstrated superior antimicrobial properties [65]. Volatile antimicrobials are also effective at penetrating most foods without direct contact, as seen in the ground beef box [66]. 

### 3.3. Active Packaging Incorporating Gas Scavengers

Using various techniques such as oxygen scavengers, carbon dioxide absorbers and emitters, and ethylene scavengers, active packaging improves the package’s internal condition for shelf life extension and improved consistency and protection [67,68]. Since the oxygen and carbon dioxide indicators are located within the box, they are classified as internal indicators. Commercial oxygen scavengers introduce chemical concepts such as iron powder oxidation, ascorbic acid oxidation, and photosensitive dye oxidation, which are easily visible by color changes after oxidation [53].

It can be used alone or in conjunction with changed atmosphere packaging. Carbon dioxide may be applied to the packaging to prevent microbial growth and reduce the rate of respiration of some products, or carbon dioxide scavengers may be used to extract carbon dioxide from freshly roasted coffees [53,69].

### 3.4. Smart and Intelligent Packaging

Against all expectations, the projected increase in packaged product innovations, the global packaged food market will jump to $26.7 billion by 2024, which holds plenty of future promises based on digitization [process of converting information into a digital (i.e., computer-readable) format. The result is the representation of an object, image, sound, document, or signal (usually an analog signal) by generating a series of numbers that describe a discrete set of points or samples] and has the vast potential [70]. Development of sensors and materials will lead to revolutionary package and system designs, which will then require additional study on protection, product shelf life, marketability, and, as well as various customer and industrial demands, for which future new packaging and design alternatives will be created.

## 4. Intelligent Medicine Packaging

The desire to be omnipresent is the most influential characteristic of technology. Patients who are elderly or have chronic illnesses cannot obey their doctor’s orders, resulting in prescription noncompliance. Prescription misuse is often undetected by doctors, and it can lead to serious and life-threatening complications. In this case, an intelligent drug packaging system would require the hour to dispense, remind patients, and monitor the medication that has been given [71]. As a result, RFID technology has grown in popularity in recent years to meet all of these increasing demands [72]. In an ideal scenario, it should reach every item in the complete framework [73]. The system should be configured to monitor each drug packet, record each tablet’s therapeutic function, and provide the patient with complete medication information [74]. 

The packaging for the smart system should be intelligent and interactive. It should be able to naturally execute all of the advantages of IoT while still being reasonably priced. In light of this, interactive and intelligent medication packaging (Figure 4) is an appropriate product that meets all of the requirements [75]. Communication, sensing, show, RFID, and other essential packaging system functions are integrated into this medicine packaging [72]. There is an effective way to provide a synergy between the display and paper-based actuators in delivering consumers with on hand or off-site to know when something exciting is going on and warn others of it [76]. This packaging will result in path-breaking transformations in the corporate world. 

Conventional transformation of information has been static, and that would change into dynamic [77]. The conventional knowledge flow from the manufacturer to the consumer was unidirectional. It will be two-way, from the consumer to the manufacturer and from the customer [78]. The packaging’s function, previously passive and controlled by the consumer, would change to active, allowing it to be self-controlled or controlled remotely. Packaging, which used to be solely responsible for containing and securing products, can now be granted additional duties and functions [79]. It serves as a contact channel between the consumer and the manufacturer or supplier. It may act as a presenter of information, a collector of information, an on-site seller, and perform some operations. 

Regarding health-IoT applications, touch sensing may be used to access intelligent and interactive packaging electrically. As a consequence, it provides solutions to the control of preventive medicine [75]. The iMedBox sends commands to the appropriate iMedPack, which responds with details on the medication slots that have been opened and are still intact. As a result, patients are given prompt reminders to prevent drug abuse [80]. Figure 4 depicts a standard implementation scenario for the Health-IoT scheme. The system’s heart is a strong, intelligent medicine box (iMedBox), which serves as a “medication inspector” and a “on-site investigator” in regular monitoring as well as a conventional in-home medicine container (such as a drawer of cabinet, a thermostat, or an icebox).

On the one hand, it is connected to the public through wireless internet (e.g., the hospital, the medicine supply chain, and the emergency help center). On the other hand, it uses radio frequency identification (RFID) links and a wireless biomedical sensor network (WBSN) to monitor a set of intelligent pharmaceutical packages (iPackage) and a set of wearable biomedical sensor tags (iTag). More information can be found in our previous work [80]. 

Furthermore, existing processes seldom incorporate new products or employ new manufacturing techniques, which are essential components for introducing new devices or solutions to the healthcare sector. This paper proposes an intelligent home-centered healthcare IoT scheme, iHome Health-IoT, based on the issues mentioned above. The iHome Health-IoT System is depicted in Figure 4 as a model. As a home healthcare gateway, an intelligent medicine box (iMedBox) is used. Wearable sensors and intelligent medication packaging (iMedPack) are just two examples of IoT devices that are seamlessly linked to the iMedBox through a heterogeneous network that is compliant with multiple wireless technologies. The Bio-Patch, which is worn on the body, can detect and relay bio-signals to the iMedBox in real-time. The iMedPack is linked to the iMedBox via an RFID connection to help users manage their prescription medications. On the iMedBox, all of the collected data is interpreted, stored, and displayed locally. The processed data can also be sent to the Health-IoT network for further study or clinical diagnosis.

## 5. Structural Design of the iMedBox-Future Vision

The system is designed by integrating multiple interfaces in the iMedBox [81]. These include the touch screen, the liquid crystal display (LCD), a microphone, speakers, light-emitting diode (LED), vibrators, and a camera [82]. The following are the distinguishing characteristics: for networking, database access, and knowledge sharing, it can be linked to a WiFi or 4G/5G communication network [83]. It’s also connected to a global positioning system for assistance that can be sent to a specific location. The WBSN function aids data collection and star-topology networking [84]. The RFID function enables the management of medicine inventory, enforcement, and the monitoring of controlled delamination material (CDM). 

The iMedBox’s architecture differs from traditional packages in that it includes control circuits and CDM films. The CDM film comprises a three-layered foil with an aluminum top sheet, an electrochemical epoxy with adhesive properties in the middle layer, and an aluminum bottom layer [85]. 

An electrochemical reaction occurs when the voltage difference between the top and bottom layers is greater than the threshold voltage. During the manufacturing of the material, it is critical to consider particular time and voltage parameters, otherwise, the epoxy layer can shatter [86]. In noncompliance, the CDM’s function is to prevent the use of intelligent and interactive packaging. The adhesive glue on the CDM’s middle layer can only be reached by sending a command to the intelligent package’s microchip [87]. As a result, CDM has a dedicated section that is managed separately for each tablet. As a result, the intelligent package has complete control over each capsule [88]. Consequently, tablet by tablet compliance is achieved as per the prescription (Figure 5).

The energy needed to open the CDM could be delivered wirelessly via near-field magnetic resonance or self-powered via a battery. Both of these approaches are modern and will produce the desired results [87]. Interactive technologies (iTAG) comprises of a wireless body sensor networks (WBSN) interface, biomedical sensors, batteries, and a low-power microcontroller (MCU) [84]. As compared to self-powered and battery-free devices, this device has incredible sensing capabilities and communication range. 

This paradigm shift has three aspects, as seen in Figure 6.

(1) Healthcare information systems (HIS) inter-organizational implementation: The information systems (ISs) [87] of all stakeholders participating in the Health-IoT production chain form the backbone of the iHome Health-IoT framework. Cloud computing [87] has provided a viable environment for such inter-organizational convergence against the so-called Health-IoT-in-Cloud.

(2) HIS cross-border development: The iHome Health-IoT system’s in-home terminal, iMedBox, serves as a connection between in-home healthcare devices and the HIS. The stakeholders’ ISs can be efficiently expanded to a patient’s home by installing individual applications in the iMedBox. As a result, the IHIS exists in the form of both businesses and consumers’ homes as the so-called Health-IoT-at-Home. This is in line with the previously stated pattern of medical services shifting from hospital-based to home-based [85].

(3) HIS personalization: Personalized programs will be the predominant method of healthcare in the future. Wearable biomedical applications with ultra-low power and low cost, such as Bio-Patch, allow personalized HIS access to patients’ bodies, resulting in the so-called Health-IoT-on-Body. As a result, HIS knowledge and contact can be handled at the level of a single individual’s body.

We created the Health-IoT platform with the current and future relevance of IHIS and IoT in the e-health sector in mind. It can be used in patient’s homes and nursing homes. The proposed device combines SoC technology, material technology, and advanced printing technology to create a patient-centric, self-assisted, fully automated, intelligent in-home healthcare solution. Environmental surveillance, vital sign acquisition, drug control, and healthcare facilities are only a few cases where the established functions can be used. The creation of several innovative, intelligent devices, including Bio-Patch, iMedPack, and iMedBox, to realize the vision of iHome Health-IoT is described in this article.

## 6. Unobtrusive Bio-Sensor

The primary goal of accurate detection is to allow continuous monitoring of actual exercises and behaviors and physiological and biochemical boundaries in the subject’s daily life. ECG, ballistocardiogram (BCG), pulse, circulatory strain (BP), blood oxygen immersion (SpO2), center/surface internal heat level, conduct, and actual exercises are the most commonly estimated imperative indications [88,89,90]. Unpretentious detection can be carried out in two ways: (1) sensors are worn by the subject, such as shoes, eyeglasses, earring, clothing, gloves, and watches; (2) sensors are mounted in the surrounding climate or as smart objects associated with the subject, such as a bench, seat [91,92], vehicle seat [93], sleeping pad [94], reflect [95], controlling wheel [96], mouse [97], latrine seat [98] and restroom scale [99]. 

A mobile phone can collect data and transmit it to a remote location for capacity and analysis. In the adjacent section, we’ll discuss some unobtrusive detecting techniques for securing fundamental signals. Figure 7 depicts the various impact classifications in a diagrammatic representation and associates organ action with key characters [100].

### 6.1. Capacitive Sensing Method

Normally, a capacitance-coupled detecting technique is used to estimate biopotentials such as electroencephalogram (EEG) and electromyogram (EMG) [101]. The skin and the terminal structure are the two layers of a capacitor in this technique. A few problems, such as skin disease and sign weakening, which are caused by glue anodes in long haul checking, can be avoided by avoiding direct contact with the body. Table 1 summarizes some popular applications of capacitive ECG detection; capacitive detecting can also be used for other applications, such as respiratory estimation, for example, by integrating a capacitive material power sensor into clothing [102], or a capacitive electrical field sensor cluster put under a dozing sleeping pad [103]. It may result in a low flag clamor percentage, causing challenges in the frontend simple circuit plan. The speaker’s information impedance must be massive to reduce the shunt influence bound by the capacitor and the information impedance (>1 T^ω^). In noncontact detecting, movement antiquities may also prove to be significant. Recently, a few solutions have been presented to address these difficulties and achieve robust estimation in real-world situations. For example, Ache et al. gradiometric estimating approach can greatly reduce movement antiques [104].

### 6.2. Photoplethysmographic Sensing Method

The photoplethysmographic (PPG) technique, which utilizes a light source to discharge light into tissue and a picture locator to collect light reflected from or transmitted through the tissue, has been widely used to estimate a variety of essential indications, including SpO_2_, pulse, breathing rate, and blood pressure. The feature estimated by this technique corresponds to the pulsatile blood volume fluctuations in the peripheral microvasculature induced by each heartbeat. Typically, the detecting unit is placed directly on the skin. The ongoing investigation has revealed that sensors may be included in commonplace wearables or contraptions such as an earring, a glove, or a cap to enable direct estimation. Table 2 summarizes the numerous types of PPG estimation gadgets used at various locations on the body. Jae et al. [105] proposed a backhanded contact sensor for estimating PPGs while wearing clothing. A control circuit was incorporated to adjust the light power for various types of garments. Similarly, Poh et al. [106] demonstrated that pulse and breath rate could be determined from PPG captured from a subject’s face using a simple computerized camera. 

### 6.3. Model-Based Cuffless B.P. Measurement

A sphygmomanometer, which has been used to estimate blood pressure for more than a century, is based on an inflatable sleeve. Conventional procedures such as auscultatory, oscillometric, and volume clip are ineffective for measuring modest blood pressure. The pulse-wave proliferation strategy is a potential technique for noninvasive blood pressure estimation. It is dependent on the relationship between the beat wave velocity (PWV) and the blood vessel pressure in accordance with the Moens–Korteweg criterion. Heartbeat travel time (PTT), the complement of PWV, can be obtained quickly and subtly from PPG and ECG. For cuffless BP assessment, different straight and nonlinear models that transmitted BP as far as PTT have been developed. When applying this methodology for BP estimation, the subject-subordinate boundaries in the BP-PTT model should be resolved first. A simple method for achieving perfect alignment is to employ the hydrostatic weight strategy, which requires patients to hoist their hands to specific heights above/below the heart level [107].

A hypothetical link between PTT, BP, and stature can be expressed in Poon and Zhang [108], this model-based cuffless BP estimation methodology can be updated at a variety of phases for uncomplicated verification. While the precision of this methodology has been validated in multiple ongoing studies [90], there are concerns about its use as a substitute for BP estimation. The key perplexing variables in the existing association between BP and PTT have been identified as vasomotor tone and pre-launch period [109]. In the future, new models should be developed to recall these vexing aspects of the request to evaluate the capability of a PTT-based method for unadorned BP estimates.

## 7. Other Types of Unobtrusive Sensing Methods

Typically, strain sensors were used to detect body movement, such as breath, heart sound, and BCG. A piezoelectric link sensor with a piezoelectric polymer as the detecting component has been used to monitor breath rate [110]. Adaptable and delicate sensors, such as piezoresistive texture sensors [111] and film-based sensors such as polyvinylidene fluoride (PVDF) and electromechanical film (EMFi) sensors [112], have been widely used in cardiac applications due to their simplicity of integration into dresses or ordinary objects such as a seat or bed. A comparative evaluation has been conducted to evaluate the two separate strain sensors’ performances in estimating body movement [113]. They indicated that only modest differences in pulse estimation between PVDF- and EMFi-based sensors were seen, notably at the prostrate position, possibly because of their differing sensitivities to various power components.

Inductive/impedance plethysmography is another widely used technique for estimating respiratory capacity that has been developed in the form of garments and material belts [113,114]. Two sinusoidal wire curls located at the rib confine and mid-region are powered by a current source that generates sinusoidal current with high repetition. The expansion of the chest during inhalation alters the inductance of the coils, and so controls the sufficiency of the sinusoidal current from which the respiratory sign can be demodulated. A continuing report discussed the exhibitions of four various techniques for wearable breath measurement, including inductive plethysmography, impedance plethysmography, piezoresistive pneumography, and piezoelectric pneumography. It was determined that piezoelectric pneumography was the most robust technique for estimating respiratory rate [107].

Similarly, optical filaments have been implanted into everyday goods or garments for direct inspection. In comparison to electronic sensors, they are impervious to electromagnetic interference. Fiber Bragg grinding has recently been presented as a vibration sensor for estimating BCG. Due to the body vibration caused by breathing and cardiovascular withdrawals [110]. A pneumatic pad based on this method has been developed to monitor the physiological conditions of pilots and drivers [110]. D’Angelo developed an optical fiber sensor embedded in a shirt to track respiratory movement [115]. Other strategies for unobtrusive physiological assessment have been presented as well, including a recurrence balanced continuous-wave Doppler radar for BCG measuring and radiometric detection for internal heat level estimate [116].

## 8. Concept of the Flexible Bio-Patch

The widespread usage of multimedia healthcare systems, combined with recent advancements in nanotechnology, has prepared the way for more pervasive and individualized healthcare systems, such as remote health monitoring and digitalized telemedicine. The combination of nanotechnology and medicine with electronic networks has resulted in the development of a diverse array of nano biosensors for healthcare monitoring, which can accurately and reliably assess all of the body’s critical physiological signals in real-time for clinical diagnosis and analysis. All bio-signal monitoring systems used in hospitals are large, with electrodes and wires, and can only monitor a single bio-signal. Numerous attempts are being made to develop new and superior wearable-nanoscale devices capable of detecting all physiological biosignals in a single device to monitor daily life. The flexible, wearable nanosensors are a component of the fast-developing market for portable devices for point-of-care (POC) monitoring [117] (Figure 8).

The new wearable and flexible biosensors should be flexible, less cucumberlike, and capable of efficiently detecting many biological signals and establishing a robust sensor network for multichannel measurements. Wearable nanosensors are constructed using a flexible substrate coated in a conducting polymer and a sensing element (bioactive element) on which electrode patterns are printed. Additionally, a transducer is employed to convert the biological signals generated by the sensor device. The biosignals corresponding to the desired parameter, such as temperature, blood glucose level, blood pressure, heart rate, oxygen and carbon dioxide levels, are then translated programmatically to readable outputs. The most often utilized substrates for nanosensors are Bio-Patches, which are employed in various medical applications.

Bio-Patches are disc-shaped disinfectants used to prevent infection following surgery, injections, and skin injury. They cannot be utilized to make wearable nanosensors. As a result, flexible substrates such as polyamide, paper, silk, polyethylene terephthalate (PET), and polydimethylsiloxane (PDMS) are recommended [118]. The Bio-Patches that will be utilized as a substrate for nanosensors should be resistant to wear and fracture and bendable and conformable with a high degree of stretchability [119,120,121]. As a result, it would be necessary to select and produce Bio-Patch materials that are intrinsically flexible and stretchable while also possessing favorable electrical properties.

Point-of-care (POC): patient care in the emergency room, in primary clinics, at home, or in other nonhospital settings where a diagnosis can be made, and treatment can be administered. Surface-enhanced Raman scattering (SERS): a spectroscopic approach that can detect chemical and biological species down to a single molecule without labeling.

## 9. Synthetic Substrate Materials

Synthetic polymers are the most frequently utilized Bio-Patch substrates because they efficiently enable low-cost, flexible designs. Due to its non-toxicity, high elasticity, chemical inertness, oxygen permeability, and thermal stability, polydimethylsiloxane (PDMS) is frequently employed in biomedical devices and nanosensors. Additionally, the electrodes for biosensing can be easily implanted due to their good conductivity [118]. Synthetic polyester fiber (PET), is a suitable substitute for silicon Bio-Patches due to its inertness, low cost, and mechanical qualities [122]. Additionally, it is molded into various patterns and shapes using ultra-thin films to provide high-contact surfaces with optical transparency [123]. Due to its chemical, thermooxidative, and ultraviolet resistance, polyethylene naphthalate (PEN), a synthetic polyester, also functions as a good plastic substrate [124].

Additionally, it possesses high inherent flexibility and bendability despite being more rigid than PET. PET and PEN both demonstrate outstanding adhesion properties for functional components such as nanoparticles, metal oxides, and conducting elements in order to construct thin structures [125]. Additionally, the screen printing technique has simplified the assimilation of microelectrodes and other microdevices [126]. Polyamides (PI) are also attractive substrates due to their outstanding dielectric and structural stability, tensile strength, and biocompatibility [127]. Paper is a low-cost substrate that can be employed directly or conformed into paper-based structures to serve as a Bio-Patch for a range of sensing modalities such as optical, electrochemical, and electrical sensing to detect bio-targets [128]. Paper is easily turned into composites due to its thin and porous structure. Additionally, it facilitates the fabrication of paper-based electrodes through processing modalities like screen printing, inkjet printing, and nanopatterning [129]. Because paper has a low tensile strength and is easily torn, the bioactive report is created by combining paper with different biomaterials. For instance, cellulose fibers and nanofibers (CNF) are reinforced with polymers such as glyoxylate polyacrylamides (GPAM) and/or polyamide epichlorohydrin (PAE) [130].

Textiles and fibers like wool, cotton, and synthetic fibers (nylon, polyester, etc.) all serve as intriguing novel materials for usage as Bio-Patches in nanosensors. Additionally, they can be easily integrated with conductive materials via a variety of manufacturing processes. Additionally, they exhibit a high degree of durability, stability, and inherent flexibility [131]. Gelatine, silk proteins, and polysaccharides have all been proposed as Bio-Patches for flexible nanosensors. Silk is a naturally occurring polymer whose proteins can be synthesized in gels, fibers, sheets, and films [132]. As a textile, it is physically resistant and provides optical transparency in the form of thin and ultrathin sheets [133,134]. Conducting polymer inks are incorporated into the films to create flexible biosensors [135]. Additionally, cellulose and chitin have been employed as substrates and components in electrochemical biosensors [136].

A fascinating technique for reducing the flexibility of Bio-Patches and devices is to employ specialized nanoscale architectures or designs that promote bendability and stretchability [137,138,139]. Nanofabrication developments have permitted the transformation of stiff and highly rigid inorganic and organic substances such as quartz, silicones, and metals into ultrathin films and circuits with ultrathin (nanoscale) structures. Cutting, folding, and buckling processes are used to manufacture flexible electronics and nanoscale biosensors [140]. All of these mechanical deformation techniques (i.e., rolling, twisting, and bending) preserve the intrinsic properties of the Bio-Patches by utilizing highly conductive materials such as gold, copper, and silver (Au, Cu, Ag) [141].

As with other biosensors, even the flexible nanosensors incorporate sensing elements such as biological components coupled with transducers into the Bio-Patch, which serves as the substrate in the case of tiny flexible nanosensors for the detection of specific analytic for the quantitative measurement of biochemical parameters such as enzymes. The flexible Bio-Patches must be integrated with the biomolecules to prevent delamination and detachment of the biomolecules from the Bio-Patch during analyte interaction and ensure their life and proper function [142].

Additionally, the Bio-Patch should be compatible with the biological sensing element, as this determines the nanosensor’s specificity and sensitivity. Additionally, the biomolecule used as a sensing element for analytic detection must be stable enough to remain attached to the Bio-Patch substrate even when subjected to mechanical shocks. Covalent and non-covalent bonding using various linkage chemistries can be used to immobilize the biological sensing element onto the Bio-Patch substrate [143]. At the moment, enzymes are the most popular bio-receptors/biomolecules as sensing components in flexible nano-biosensors due to their application in electrochemical detection, low cost, high specificity, and high affinity [144]. Due to their great sensitivity, mobility, and dependability, flexible nano biosensors generate electronic and electrochemical signals to the transducer and, to a lesser extent, optical signals [145]. Additionally, electrodes may be easily integrated into Bio-Patches using particular techniques used in flexible nano-biosensors.

## 10. Flexible Conducting Polymers

Conductivity is accomplished in the nanosensors by utilizing polymers with redox activity, doping, and mixed electronic and ionic transport. By altering conductive polymers’ physical and chemical properties, doping increases the efficiency and processability of conducting polymers by several orders of magnitude. Conducting polymers entrap the biological sensing elements either physically or chemically, enabling the integration of biomolecules with conducting polymers and Bio-Patch substrates for bio-signal detection in a nano-biosensor. Polyethylene dioxythiophene (PEDOT), is one of the most extensively investigated conductive polymers (3,4-ethylene dioxythiophene). It exhibits superior electrical stability and is extremely conductive. A PEDOT:PSS conjugate is a superior flexible polymer electrode for sensing glucose, DNA/RNA, and biomarkers [146,147]. Metals such as Au and Ag are promising choices for constructing flexible electrodes because they are ductile and can be shaped into thin films. They can also work in conjunction with the biomolecules employed in the sensors and can withstand mechanical deformations while retaining their electrical conductivity (e.g.; mesh, serpentine, wavy, etc.) and other carbon nanostructures, such as carbon nanofibers (CNF) and carbon nanotubes (C60), have also found use in flexible biosensing technologies [148].

## 11. Flexible Bio-Patch Nano-Sensors

The flexible, wearable Bio-Patch nanosensors attach to biological surfaces such as skin and tissues to assess physiological signals such as breath, blood pressure, temperature, stress, pulse rate, and biochemical signals such as glucose and oxygen uric acid, and dopamine levels. As previously discussed, various synthetic substrates can be employed as flexible Bio-Patches onto which nanosensors can be incorporated in order to detect a desired biological or physiological signal. The Bio-Patch prototype consisted of a pair of electrodes printed on a piece of photo paper, a SoC (system-on-chip) sensor, and a depleted battery. The nano-chip is integrated with conducting elements such as metal nanostructures (e.g., Ag, Au, etc.) and is created onto the flexible Bio-Patch. The Bio-Patch nanosensor is applied to the skin surface and detects essential physiological data. Biosensors detect and convert biological signals from the body into measurable signals such as electrochemical, optical, or piezoelectric signals. Following that, these signals are examined and processed programmatically via a wireless network. Doctors and patients can obtain the results digitally. A Bio-Patch can operate independently or in conjunction with other Bio-Patches via an active cable printed on a substrate for recording multichannel bio-signals [149] (Figure 9).

### Applications of Flexible Bio-Patch Nano-Sensors

Flexible Bio-Patch nanosensors are designed solely to provide tailored and ubiquitous e-healthcare monitoring to acquire continuous information about a patient’s physiological and biochemical status. Additionally, the bio-patches are produced in such a way that they support the flexible healthcare device’s symbiotic function. Being the body’s most accessible organ, the skin offers a large surface area for the application of flexible Bio-Patch nanosensors. All physiological and biochemical parameters can be monitored and identified by attaching flexible nanosensor patches to the skin. The following are a few of the applications (Table 3): (a) to record ECG (electrocardiogram) and EMG (electromyogram) signals; further, the cardiac health sensor was designed to monitor bio-impedance in addition to ECG; (b) human stress monitoring patch made of perforated polyamide membrane integrated with sensors for skin temperature, skin conductance, and a flexible pulse wave sensor to detect the pulse wave via multimodal biological signals; (c) flexible nanosensors can also be used to detect pathogenic organisms such as human immunodeficiency virus (HIV) and *Escherichia coli* (*E. coli*); (d) apart from its use as flexible and wearable nanosensors, flexible bio-patches have been identified as promising candidates for drug and vaccine administration and wound healing. Disposable photovoltaic patches were designed to deliver electric stimulation (ES), increase regenerative activity, and aid in wound healing. Thus, flexible bio-patches offer a plethora of applications in nanosensors and medicine.

## 12. Future Perspectives towards the Fusion of Nanotechnology with Biosensors

The food sector has achieved great strides in recent years in packaging innovation through the development of new functional products, transport and controlled release of bioactive compounds, and pathogen and pesticide detection with nanosensors and indicators. Intelligent packaging is still a relatively new concept in food packaging (it is still in the developing stage); when progress is achieved, it is critical to consider which features to emphasize as underlying objects for effective goods. However, it paves the way for a bright future in terms of food safety and presents a hopeful image for years to come in terms of developing revolutionary technologies such as nanotechnology. Although the potential applications and benefits of nanotechnology are tremendous, their implementation in the food industry is comparatively new compared to their usage in drug delivery and medicines. Nanotechnology can completely transform the global food system by developing new food products, improving packaging and storage techniques, and modifying the fundamental functionality of food.

Additionally, intelligent food packaging that incorporates nanosensors may give consumers with information about the state of the food contained within. Food packaging contains nanoparticles (NPs) that notify consumers when a food is no longer safe to ingest. Nanotechnology advancements will very certainly alter the manufacturing process for the entire packaging business.

Nonetheless, less work has been directed into food-related nanotechnology applications, and the majority of them require a high level of research and development to ensure their safety. Ongoing research involves using sensors to transmit required information and stimulate subsequent changes in packaging materials, surroundings, or products to ensure their preservation and safety. There are numerous obstacles to the future success of nanotechnology in food packaging, public approval, economics, and regulation of food treated with nanoparticles that could accumulate and cause toxicity. The safety of NPs in the food business is another issue that both government and industry must address. Neither is it certain that nanotechnology is completely safe or hazardous to health. The food processing sector must instill consumer confidence and acceptance in the safety of nanoparticle foods. Although there is considerable evidence supporting food-grade nanomaterial (NMs), there are reports that support its toxicity, or it has not been examined yet. Therefore, when it comes to industrial-scale deployment of nanotechnology, it is critical to measure the release of nanoparticles into the environment and estimate subsequent exposure levels to these materials, as nanoparticles can penetrate human organs and organelles. The duration of exposure, exposure concentrations, penetration sites, immunological response, accumulation and retention of N.P.s in the body, as well as their future consequences, should all be carefully examined. As a result, mandatory testing of nano-modified foods should be conducted prior to their commercialization.

Additionally, there will be a strong reliance on governments, regulatory bodies, and manufacturers to address the concerns above. In the long run, nanoscience and nanotechnology applied to food may develop into a new field of research dubbed ‘food nanotechnology’. In addition, wearable biosensors are predicted to become more simplified in the next years, moving away from the wrist and into fabrics and fashion accessories that integrate seamlessly with the wearer’s daily life. To overcome fouling difficulties, several of these devices will require disposable components. Future wearable biosensors will enable the noninvasive monitoring of a wide variety of biomarkers (including proteins and nucleic acids), enabling complete medical diagnoses and performance assessment. Acceptance of these noninvasive biosensors by the medical community will require substantial and successful validation in human trials and a better comprehension of the sensor information’s therapeutic relevance. We foresee exciting new advances in the near future, given the competitive research and significant commercial prospects in wearable biosensors. Thus, the market for wearable sensors is projected to continue its rapid expansion and its trajectory of transforming and improving people’s lives.

## 13. Conclusions

The food and pharmaceutical processing industries, in particular, have benefited significantly from the current development of nanotechnology. Intelligent packaging systems are a fast emerging industry that will prioritize food security in the coming years. The advancement of nanosensor technology, the inclusion of nanosensors into food packaging, and the creation of breakthroughs in intelligent packaging (IP) solutions are all critical for advancing food security. This innovative packaging system will assist in identifying, managing, analyzing, documenting, and communicating the supply chain. The connectivity of nanosensors expands a single nanosensor’s capabilities by allowing it to collaborate and exchange information; as a result, energy optimization coding for wireless nanosensors (WNSNs) will have a significant impact on practically every element of our civilization and will affect our daily lives. These networks are still in their infancy; one example is the disadvantages of nano communication and nanobatteries. The development of printed electronics for mass production is directly tied to the commercialization of these developing technologies; smart labeling and packaging are predicted to be less expensive than food product. Research into these smart packaging technologies may result in system upgrades. In the future, smart packaging offers untapped potential to provide customers with benefits and comfort.

## Figures and Tables

**Figure 1 nanomaterials-11-01515-f001:**
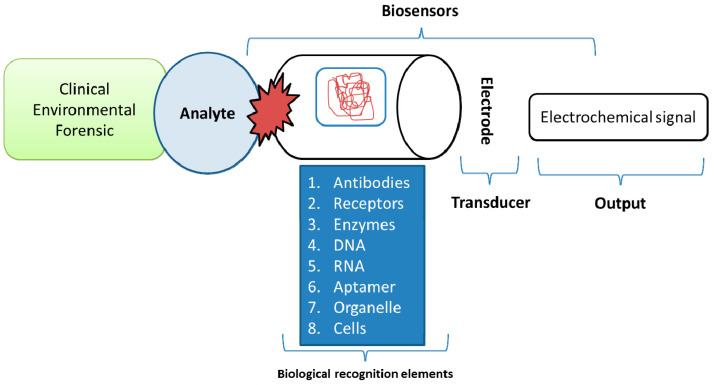
Indicates the fundamental theory of biosensor.

**Figure 2 nanomaterials-11-01515-f002:**
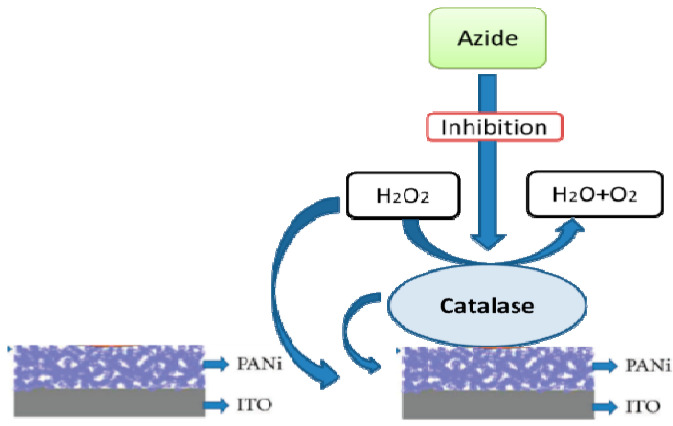
This is a diagrammatic illustration of a bio-electrode to improve the bio-sensor for the hydrogen peroxide grit in an optimal condition. (Singh et al. [2] (Creative Commons Attribution License, which permits unrestricted use, distribution, and reproduction in any medium, provided the original work is properly cited) described a disposable biosensor capable of rapidly detecting not only H_2_O_2_ but also azide in biological samples utilizing a CAT/PANi/ITO electrode as a bioelectrode. This film is highly effective at keeping enzyme activity and preventing it from escaping the film. This indicates that this effective film can be used to immobilize not only catalase but also other enzymes and bioactive chemicals, making it a potentially useful platform for the development of biosensors).

**Figure 3 nanomaterials-11-01515-f003:**
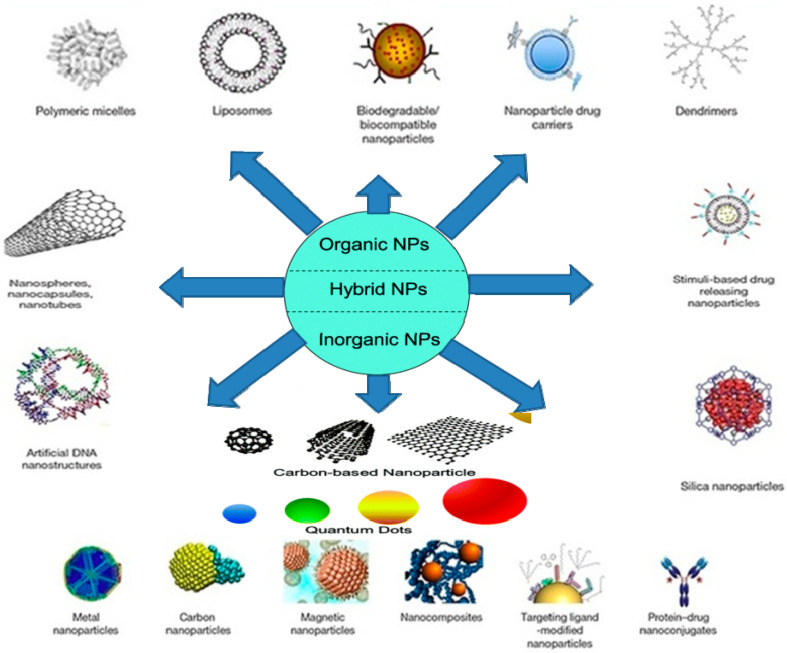
Schematic illustration of different types of nanomaterials used in biomedicine.

**Figure 4 nanomaterials-11-01515-f004:**
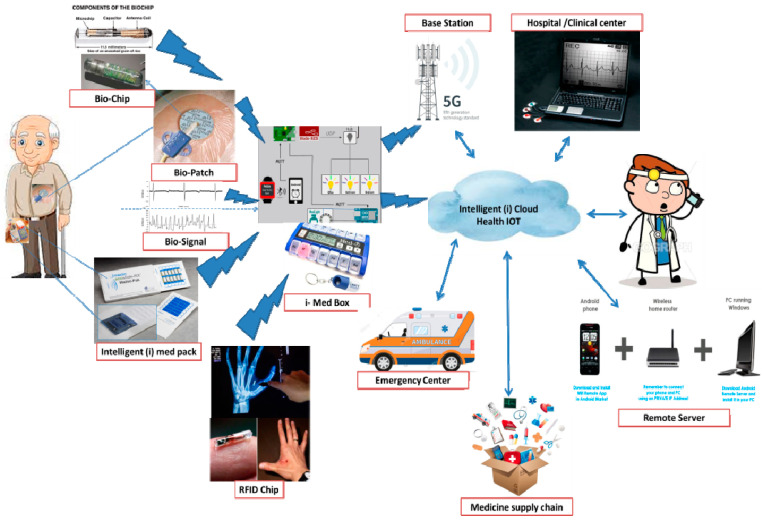
Application scenario for the proposed iHome Health-IoT system.

**Figure 5 nanomaterials-11-01515-f005:**
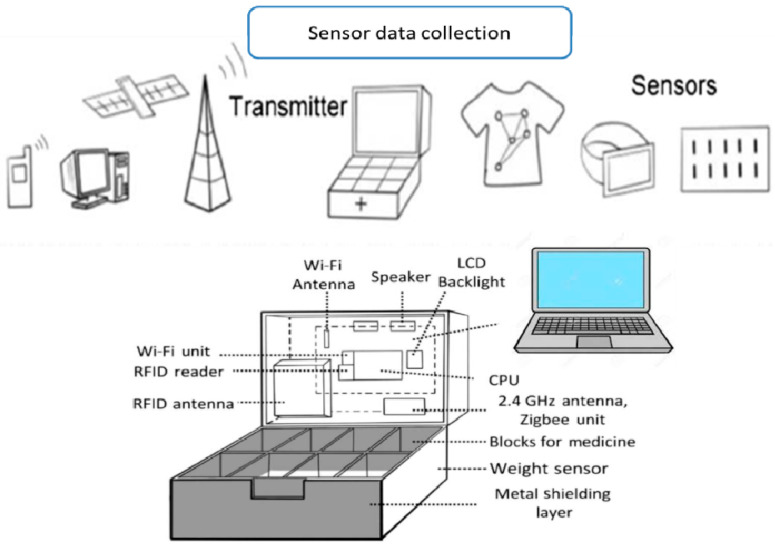
Architectural design of the iMedBox.

**Figure 6 nanomaterials-11-01515-f006:**
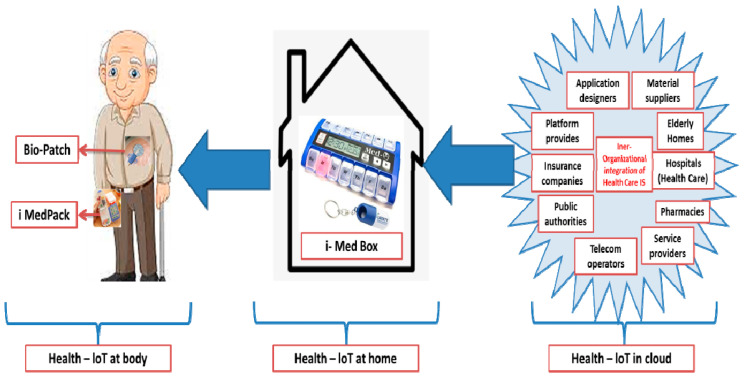
Integrated HIS enabled by the proposed iHome Health-IoT system.

**Figure 7 nanomaterials-11-01515-f007:**
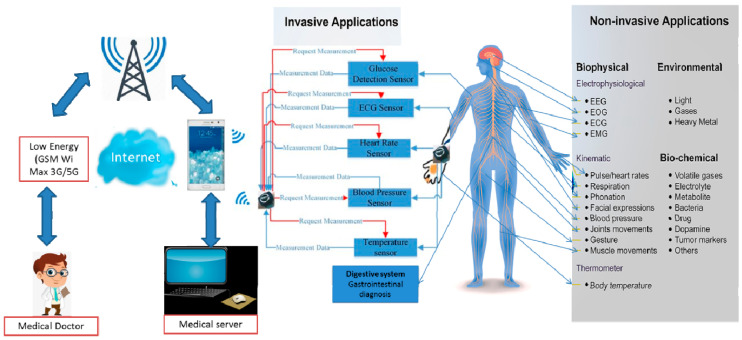
Unobtrusive biosensor impact classification and health monitoring system. (Point-of-care(POC) diagnostics play an essential role in delivering health care, particularly for clinical management in different organ and surveillance). Brief sensor platform for health monitoring. A significant distinction can be made between noninvasive and invasive applications, including wearable sensors for monitoring biophysical, biochemical, environmental signals, and implantable devices for nervous, cardiovascular, digestive, locomotors system.

**Figure 8 nanomaterials-11-01515-f008:**
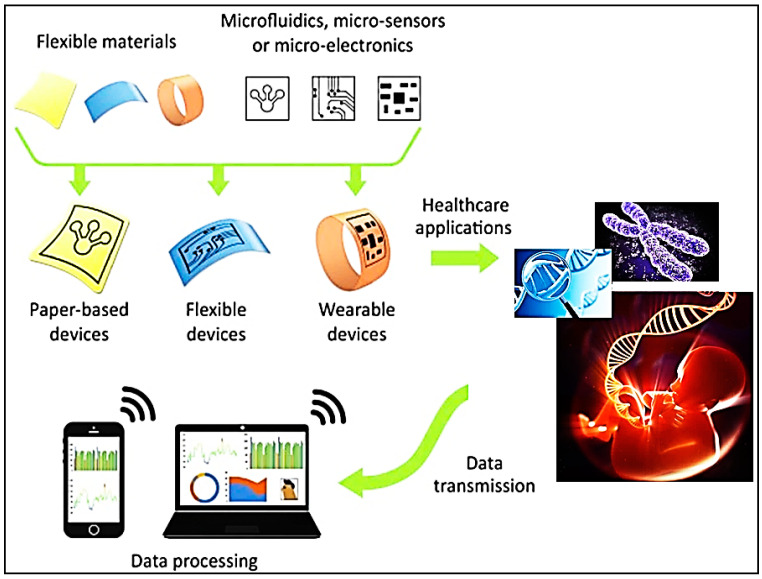
Overview on the working of flexible devices for POC diagnosis. (We discuss recent advancements in the multiplexing capabilities and sensitivity of paper-based POC diagnostics, and then discuss the development of flexible polymer-based POC diagnostics for sensing biological targets, as well as their prospective uses as wearable devices).

**Figure 9 nanomaterials-11-01515-f009:**
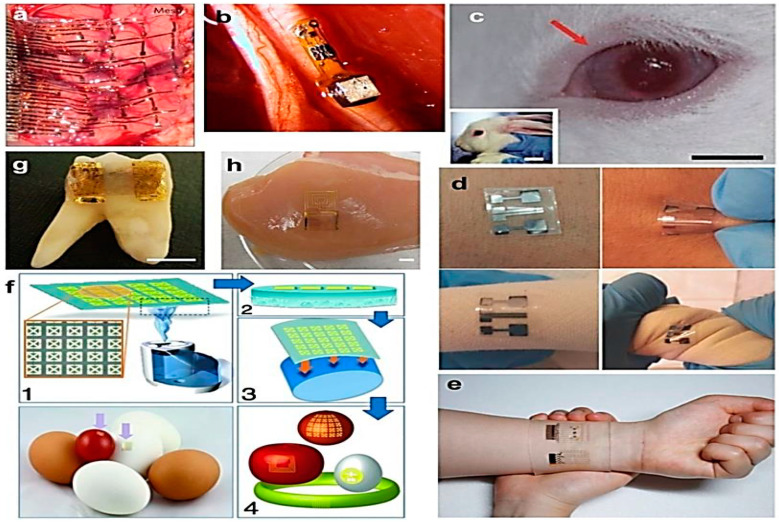
The possible locations where a flexible bio-patch nanosensor can be used [142]. (**a**) Image of fabricated sensor; the sensor can be attached to the surface of the meat or be placed onto the inside lining of the package. The status of the meat product can be monitored using smart phone by taking a photo of the sensor. (**b**) Schematic representation of self-powered multifunctional electronic skin used for continuous monitoring of lactate, glucose, uric acid, and urea in exercise-induced sweat using piezoelectric-linked enzymatic biosensors. During exercise, this device functions without an additional power supply through piezoelectric–enzymatic reaction coupling. (**c**) A wireless glucose sensor incorporated into a contact lens platform with wireless power transfer circuitry and display pixels for a fully integrated and transparent platform that does not hinder vision. This device detects fluctuating tear glucose concentrations through a resistance-based enzymatic mechanism demonstrated in a rabbit model. (**d**) Epidermal reverse iontophoretic tattoo-based glucose sensor configuration and operation principle, with picture of device applied to a human subject. Proof-of-concept demonstration of reverse iontophoretic tattoo-based ISF glucose sensor. This indicates aa working electrode and reference/counter electrode, respectively. (**e**) Iontophoretic paper battery and skin-like biosensor for noninvasive blood glucose monitoring, applied to a human subject. The inclusion of hyaluronic acid, facilitates enhanced ISF extraction for increased ISF glucose sampling reliability. (**f**) Schematic of steps for rapid transfer of silk antennas onto curved substrates: (1) Water vapor is applied to the back of silk films, yielding (2) a film in which the back surface of the film has been partially melted. (3) This melted surface is conformally applied to arbitrary surfaces, yielding (4) applied functional sensors on various surfaces. Photos of THz split-ring resonators (SRRs) fabricated on the silk substrate wrapped on an apple. (**g**) Mouthguard-based sensor for glucose monitoring in saliva with on-body application and analysis of increasing glucose concentrations. Fully integrated saliva glucose sensor aims toward continuous in-mouth glucose monitoring. (**h**) Various demonstrations of biosensors in the food industry. (**a**) Geometric barcode sensor for monitoring chicken spoilage under different temperature conditions.

**Table 1 nanomaterials-11-01515-t001:** Various usage of capacitive E.C.G. sensors.

Systems	Area of E.C.G. Electrodes	Measured Signals and Parameters
BP monitoring chair	Chair pad and arms	ECG, PPG, HR, BP
Noncontact chair-based system	Chair back	ECG, PPG, HR, BP
Sleeping bed	Bed pad	ECG, PPG, HR, BP
Wearable ECG systems	Cloth and Bed	ECC
Aachen smart chair	Chair backrest and pad	ECC
Ambulatory E.C.G. monitoring over cloth	Integrated on underwear	ECC
Textile integrated long-term ECG monitor	Incorporated into a piece of clothing	ECG
Noncontact ECG/EEG electrodes	Implanted inside texture and attire	EEG, ECG
Remote wearable ECG sensor	Unified into a cotton T-shirt	ECG

**Table 2 nanomaterials-11-01515-t002:** PPG estimating gadgets at various destinations of the body.

Devices	Area of Sensor/Operation Mode	Measured Parameters
PPG ring	Finger/reflective	Heart rate variability, SpO_2_
Pulsar	External ear cartilage/reflective	Heart rate
Fore-head mounted sensor	Forehead/reflective	SpO_2_
e-AR	Posterior and inferior auricular/reflective	Heart rate
IN-MONIT system	Auditory canal/reflective	Heart activity and heart rate
Glove and hat based sensor	Finger and forehead/reflective	Heart rate and pulse wave transit time
Ear-worn monitor	Superior auricular/reflective	Heart rate
Headset	Ear lobe/transmissive	Heart rate
Heart-phone	Auditory canal/reflective	Heart rate
Magnetic earing sensor	Ear-lobe/reflective	Heart rate
Eyeglasses	Nose bridge/reflective	Heart rate and pulse transit time
Ear-worn PPG sensor	Ear-lobe/reflective	Heart rate
Smartphone	Finger/reflective	Heart rate

**Table 3 nanomaterials-11-01515-t003:** The list of the applications of flexible bio patches other than an anchor/substrate to the flexible nanosensors.

S. No	Type of Bio-Patch Used	Application of Bio-Patch Sensor
a	Disposable photo voltaic patches	Provides electric stimulation and promotes regeneration for wound healing.
	Tenocyte cells seeded collagen patches	Promotes healing of anterior cruciate ligament (A.C.L.) repair.
	Si-NN PDMS patch	This micro-needle patch is used for intramuscular and intratissue nano-injection of biomolecules.
b	Micro-needle patch	Transdermal delivery of dual mineralized peptides for therapy of Type-2 diabetes mellitus.
	Silk fibroin microneedle patches	Sustained transdermal delivery of the contraceptive hormone, levonorgestrel enclosed in microcarriers.
c	3-D printed F-GelMA hydrogel patches	Local delivery of the model nano-medicine PEGylated liposomal doxorubicin (DOX) against cancer.
d	Microneedle patches	Intradermal vaccination of poly (lactic-co-glycolic acid) (PLGA) micro-particles encapsulated antigens.

## Data Availability

Not applicable.

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
