# Peer review of "Development of Nanosensors Based Intelligent Packaging Systems: Food Quality and Medicine"

_nanomaterials, 2021, doi:10.3390/nano11061515_

Round 1

Reviewer 1 Report

nanomaterials-1193644-peer-review-v1

Development of Nano-sensors based Intelligent packaging systems: Food quality and medicine

General comments:

The authors provided an up-to-date description of nanotechnology-based sensors along with an interesting and exhaustive review of the smart packaging concepts (for different purposes: food industry and medicine). The core of the paper is well organized and well-referenced, and the presence of tables is particularly useful to the readers. Though, the introduction (and the abstract as well) does not present with clarity the outline of the paper and the full scope of this excellent review (a schematic or a table may help) but mainly presents nanotechnology-based sensors. Also, the introduction could better discuss the recent reviews written on similar topics to enhance/underline the strategy used to make this specific work. I also recommend broadening the scope of section 12 and improving the picture quality and the Figure legends.

Hereafter a detailed list of suggestions/questions/comments:

Line 28: typo (such)

Line 69: forum? The description (in the text and the legend) of Fig 2 could be improved (explain each term used, the reactions…) as well as the image quality

Line 74: the sentence could be reworded (do you mean: Biosensors are an emerging area of interdisciplinary research?)

Line 77: what do you mean by “embattled”? The following sentence seems uncomplete (words/verb missing) - could be merged with the previous one

Lines 93/94 and 107/108: could be reworded

Fig 3 shall be better introduced into the text and the legend of Fig 3 shall describe the different process/reaction steps, alternatively consider removing this Fig and replacing it with a schematic that shows the different types of nanomaterials, as listed in Lines 108-109.

Line 111: double-check if the acronym NADH is defined in the text

Line 148: toxins produced by microorganisms cause…

Line 173: polyethylene glycol (PEG)

Line 183 and 205: carbon quantum dots? (CQDs)

Line 187: Remove Ahmed before [42]

Line 309: BSN is defined previously but double-checks WBSN

Line 324: please double-check the link between ref [87] and your purpose:” ‘Triple wins’ or ‘triple faults’? Analyzing the equity implications of policy discourses on climate-smart agriculture (CSA)”

Line 330: please better explain the term iTag and the link to the referenced article [84]

Fig. 4: blurry, a schematic drawing of the system may help

Line 343/344: please double-check this sentence

Line 351/353 + Fig 5 (also blurry): this Fig is not well described in the text (the sentence could be simplified/reworded for a sake of clarity) and in the legend, e.g. color code for the arrows is not explained…

Line 368/375: what do you mean by “movement ancient rarities”? Motion-related artifacts? the term is used several times, please double check if this corresponds to terminology/notations made in [104] for instance

Line 370: omega in upper case

Line 411-415: this part shall be reformulated, by adding notably the formula derived by Zhang [108].

Line 461: have paved?

Line 480: “2” in O2 and CO2 in lower case

Fig 6: a bit blurry, please indicate the ref if it is a reprint (+ the associated reprint permission)

Line 534: could you double-check this sentence (is “decimated” the best term?)

Line 583: fullerenes C60?

Fig 7: a bit blurry, and the legend is incomplete, please describe in particular the four images at the bottom left.

Lines 614-628 vs Table 3: please double-check the consistency of this section (list in the text a, b, …h vs list of the table) it is difficult to follow you here, it is recommended to better introduce Table 3.

Section 12 (future perspectives): excellent analysis, but why did the authors limit this section to a discussion about the food industry? Is it possible to broaden the scope of this section?

Author Response

Respond to Reviewer-1 Comments

The authors provided an up-to-date description of nanotechnology-based sensors along with an interesting and exhaustive review of the smart packaging concepts (for different purposes: food industry and medicine). The core of the paper is well organized and well-referenced, and the presence of tables is particularly useful to the readers. Though, the introduction (and the abstract as well) does not present with clarity the outline of the paper and the full scope of this excellent review (a schematic or a table may help) but mainly presents nanotechnology-based sensors. Also, the introduction could better discuss the recent reviews written on similar topics to enhance/underline the strategy used to make this specific work. I also recommend broadening the scope of section 12 and improving the picture quality and the Figure legends.

The authors are really grateful for the reviewer valuable suggestion, towards formatting the manuscript in a readable format for the readers

Hereafter a detailed list of suggestions/questions/comments:

Line 28: typo (such)

As per the reviewer valuable suggestion the typographical errors were corrected

Line 69: forum? The description (in the text and the legend) of Fig 2 could be improved (explain each term used, the reactions…) as well as the image quality

As per the reviewer valuable suggestion the description were incorporated

Figure 2: It shows a diagrammatic illustration of bio-electrode to improve bio-sensor for the hy-drogen peroxide grit in optimal condition. (Singh et al. [2] described a disposable biosensor capable of rapidly detecting not only H2O2 but also azide in biological samples utilizing a CAT/PANi/ITO electrode as a bioelectrode. This film is extremely effective at keeping enzyme activity and pre-venting it from escaping the film. This indicates that this effective film can be used to immobilize not only catalase but also other enzymes and bioactive chemicals, making it a potentially useful platform for the development of biosensors).

Line 74: the sentence could be reworded (do you mean: Biosensors are an emerging area of interdisciplinary research?)

As per the reviewer valuable suggestion the sentence could be reworded

Line 77: what do you mean by “embattled”? The following sentence seems uncomplete (words/verb missing) - could be merged with the previous one

As per the reviewer valuable suggestion the sentence has been edited

Lines 93/94 and 107/108: could be reworded

As per the reviewer valuable suggestion the sentence could be reworded

Fig 3 shall be better introduced into the text and the legend of Fig 3 shall describe the different process/reaction steps, alternatively consider removing this Fig and replacing it with a schematic that shows the different types of nanomaterials, as listed in Lines 108-109.

As per the reviewer valuable suggestion the figure 3 has been edited (Figure 3: Schematic illustration of different types of nanomaterials used in biomedicine)

Line 111: double-check if the acronym NADH is defined in the text

The sentence has been edited according to reviewer’s valuable comment

Line 148: toxins produced by microorganisms cause…

The sentence has been edited according to reviewer’s valuable comment

Microorganism-produced toxins, which causes major health problem all over the world. Microorganisms produce toxins as a act of self-defense [28]. As a result, developing sensitive and fast methods to detect toxins in foods and related products is critical.

Line 173: polyethylene glycol (PEG)

As per the reviewer valuable suggestion the abbreviations has been edited

Line 183 and 205: carbon quantum dots? (CQDs)

As per the reviewer valuable suggestion the abbreviations has been edited

Line 187: Remove Ahmed before [42]

As per the reviewer valuable suggestion the sentence has been edited

Line 309: BSN is defined previously but double-checks WBSN

As per the reviewer valuable suggestion the abbreviations has been edited

[wireless biomedical sensor network (WBSN)]

Line 324: please double-check the link between ref [87] and your purpose:” ‘Triple wins’ or ‘triple faults’? Analyzing the equity implications of policy discourses on climate-smart agriculture (CSA)”

As per the reviewer valuable suggestion the reference has been edited

87.Pang, Z., Tian, J., & Chen, Q. (2014, February). Intelligent packaging and intelligent medicine box for medication man-agement towards the Internet-of-Things. In 16th international conference on advanced communication technology (pp. 352-360). IEEE

Line 330: please better explain the term iTag and the link to the referenced article [84]

As per the reviewer valuable suggestion the abbreviations has been edited

[The Interactive Technologies (iTAG)iTag comprises a of Wireless Body Sensor Networks (WBSN)]

  1. Sodhro, A. H., Pirbhulal, S., Qaraqe, M., Lohano, S., Sodhro, G. H., Junejo, N. U. R., & Luo, Z. (2018). Power control algorithms for media transmission in remote healthcare systems. IEEE Access, 6, 42384-42393

Fig. 4: blurry, a schematic drawing of the system may help

As per the reviewer valuable suggestion the figure 4 has been edited

Figure4a,b,c has been edited with higher DPI

Line 343/344: please double-check this sentence

As per the reviewer valuable suggestion the sentence has been edited

Line 351/353 + Fig 5 (also blurry): this Fig is not well described in the text (the sentence could be simplified/reworded for a sake of clarity) and in the legend, e.g. color code for the arrows is not explained…

As per the reviewer valuable suggestion the figure 5 has been edited

Figure 5: Unobtrusive biosensor impact classification and health monitoring system (Point-of-care(POC) diagnostics play an important role in delivering health care, particularly for clinical management in different organ and surveillance) Brief sensor platform for health monitoring. A major distinction can be made between non-invasive and invasive applications, including wearable sensors for monitoring biophysical, biochemical, environment signals, and implantable devices for nervous, cardiovascular, digestive, locomotors system.

Line 368/375: what do you mean by “movement ancient rarities”? Motion-related artifacts? the term is used several times, please double check if this corresponds to terminology/notations made in [104] for instance

As per the reviewer valuable suggestion the sentence has been edited

Line 370: omega in upper case

As per the reviewer valuable suggestion the sentence has been edited

Line 411-415: this part shall be reformulated, by adding notably the formula derived by Zhang [108].

As per the reviewer valuable suggestion the sentence has been edited

Line 461: have paved?

As per the reviewer valuable suggestion the sentence has been edited

Line 480: “2” in O2 and CO2 in lower case

As per the reviewer valuable suggestion the sentence has been edited

Fig 6: a bit blurry, please indicate the ref if it is a reprint (+ the associated reprint permission)

As per the reviewer valuable suggestion the figure DPI has been increased

Line 534: could you double-check this sentence (is “decimated” the best term?)

As per the reviewer valuable suggestion the sentence has been edited

Line 583: fullerenes C60?

As per the reviewer valuable suggestion the sentence has been edited

Fig 7: a bit blurry, and the legend is incomplete, please describe in particular the four images at the bottom left.

As per the reviewer valuable suggestion the figure DPI has been increased

Figure 7. The possible locations where a flexible bio-patch nanosensor can be used [142].

a Image of fabricated sensor; The sensor can be attached to the surface of the meat or be placed onto the inside lining of the package. The status of the meat product can be monitored using smart phone by taking a photo of the sensor.

b Schematic representation of self-powered multifunctional electronic skin used for continuous monitoring of lactate, glucose, uric acid and urea in exercise-induced sweat using piezoelec-tric-linked enzymatic biosensors. During exercise, this device functions without an additional power supply through piezoelectric–enzymatic reaction coupling.

c, A wireless glucose sensor incorporated into a contact lens platform with wireless power transfer circuitry and display pixels for a fully integrated and transparent platform that does not hinder vi-sion. This device detects fluctuating tear glucose concentrations through a resistance-based enzy-matic mechanism, which was demonstrated in a rabbit model.

d, Epidermal reverse iontophoretic tattoo-based glucose sensor configuration and operation prin-ciple, with picture of device applied to a human subject. Proof-of-concept demonstration of reverse iontophoretic tattoo-based ISF glucose sensor. Indicate working electrode and reference/counter electrode, respectively.

e, Iontophoretic paper battery and skin-like biosensor 3for noninvasive blood glucose monitoring, applied to a human subject. Inclusion of hyaluronic acid facilitates enhanced ISF extraction for in-creased ISF glucose sampling reliability.

f, Schematic of steps for rapid transfer of silk antennas onto curved substrates: (1) Water vapor is applied to the back of silk films, yielding (2) a film in which the back surface of the film has been partially melted. (3) This melted surface is conformally applied to arbitrary surfaces, yielding (4) applied functional sensors on a variety of surfaces. Photos of THz split ring resonators (SRRs) fab-ricated on the silk substrate wrapped on an apple.

g, Mouthguardbased sensor for glucose monitoring in saliva with on-body application and analysis of increasing glucose concentrations. Fully integrated saliva glucose sensor aims toward continuous in-mouth glucose monitoring.

h, Various demonstrations of biosensors in the food industry. (a) Geometric barcode sensor for monitoring chicken spoilage under different temperature conditions.

Lines 614-628 vs Table 3: please double-check the consistency of this section (list in the text a, b, …h vs list of the table) it is difficult to follow you here, it is recommended to better introduce Table 3.

As per the reviewer valuable suggestion the sentence and table 3 has been edited

(Table 3): a. to record ECG (Electrocardiogram) and EMG (Electromyogram) signals; further, the cardiac health sensor was designed to monitor bio-impedance in addition to ECG; b. human stress monitoring patch made of perforated Polyamide membrane integrated with sensors for skin temperature, skin conductance, and a flexible pulse wave sensor to detect the pulse wave via multimodal biological signals; c. Flexible nanosensors can also be used to detect pathogenic organisms such as human immunodeficiency virus (HIV) and Escherichia coli (E. coli); d. Apart from its use as flexible and wearable nanosensors, flexible bio-patches have been identified as promising candidates for drug and vaccine administration, as well as wound healing.

Section 12 (future perspectives): excellent analysis, but why did the authors limit this section to a discussion about the food industry? Is it possible to broaden the scope of this section?

As per the reviewer valuable suggestion the future perspectives been edited with addition information regarding the medical and health care application

(The future perspective were focused on the food application in major, because most resent advance and development was higher in medical and health care when compared with food application)

Reviewer 2 Report

Line 28, "Suh" should be "Such" it appears.

In general, the abstract is intended to provide a synopsis of the results presented in the paper as a whole. The abstract as written seems to be more of a random collection of sentences pulled from the text body and fails to summarize the results. After reading the abstract, the reader should be able to have an idea of what the work described accomplished, with key metrics of success enumerated. Please re-write.

The purpose of Figure 2 is unclear. How is the "grit", which is unmentioned in the text, improved? By the arrows? Where is the "grit"? 

The language of the introduction seems machine generated, as in a dictionary of terms related to nanotechnology were randomly selected by an algorithm to produce word salad that seems slightly coherent unless you actually try to understand it, at which point it becomes clear that the text is largely meaningless.

The language starting at line 130 starts to address the citations directly, making the purpose of the citations clear, and the writing now starts to make sense and is more clearly human written.

What work is connected to the toxin detection described in line 153? 

Melamine is compost? (line 163) If these cases of melamine contamination are well documented, then please provide citations. It should be simple to do so.

Lines 173-175 appear to be an unintentional combination of multiple sentences.

What is "it" in line 256?

What digitization? What is Industry 4.0? (line 263)

Figure 4 doesn't seem to show anything coherent. The diagrams are barely legible, the contents of the diagrams are unclear, the purpose of the 3 boxes opaque and the logic of the internal components suspect. Why is there a WBSN interface? What is an WBSN interface? It is never defined. In some cases boxes in the diagram are unconnected (battery, e.g.) and in others the purpose unclear. FSM? GPS? 

In the paper, acronyms are unevenly used and often not defined. Electrocardiogram (ECG) is sometimes E.C.G. The correct way is to omit the periods, as in ECG. Correct all acronyms.

The discussion of the iMedbox structure (line 303 onward) is nonsense and references patents for charger plugs, describes features not shown on Fig 4 and so on.

The so called iTag (line 330) has no breathtaking characteristics as described. It is not described at all.

Movement antiques? (line 375)

I'm halting reading of this.

Author Response

Respond to Reviewer-2 Comments

Line 28, "Suh" should be "Such" it appears.

As per the reviewer valuable suggestion the typographical error has been edited throughout the manuscript

In general, the abstract is intended to provide a synopsis of the results presented in the paper as a whole. The abstract as written seems to be more of a random collection of sentences pulled from the text body and fails to summarize the results. After reading the abstract, the reader should be able to have an idea of what the work described accomplished, with key metrics of success enumerated. Please re-write.

The authors are really grateful for the reviewer valuable suggestion, towards formatting the manuscript in a readable format for the readers

The purpose of Figure 2 is unclear. How is the "grit", which is unmentioned in the text, improved? By the arrows? Where is the "grit"? 

As per the reviewer valuable suggestion the description were incorporated

Figure 2: It shows a diagrammatic illustration of bio-electrode to improve bio-sensor for the hy-drogen peroxide grit in optimal condition. (Singh et al. [2] described a disposable biosensor capable of rapidly detecting not only H2O2 but also azide in biological samples utilizing a CAT/PANi/ITO electrode as a bioelectrode. This film is extremely effective at keeping enzyme activity and pre-venting it from escaping the film. This indicates that this effective film can be used to immobilize not only catalase but also other enzymes and bioactive chemicals, making it a potentially useful platform for the development of biosensors).

The language of the introduction seems machine generated, as in a dictionary of terms related to nanotechnology were randomly selected by an algorithm to produce word salad that seems slightly coherent unless you actually try to understand it, at which point it becomes clear that the text is largely meaningless.

The authors are really grateful for the reviewer valuable suggestion, entire manuscript has been re-modulated

The language starting at line 130 starts to address the citations directly, making the purpose of the citations clear, and the writing now starts to make sense and is more clearly human written.

As per the reviewer valuable suggestion the sentence has been edited

What work is connected to the toxin detection described in line 153? 

As per the reviewer valuable suggestion the sentence has been edited

Nanotechnology-based sensors for food analysis and Monitoring of Food Security

  • Toxins detection andtoxin producing pathogen detection

Melamine is compost? (line 163) If these cases of melamine contamination are well documented, then please provide citations. It should be simple to do so.

As per the reviewer valuable suggestion the sentence has been edited

Tian, Y., Chen, L., Gao, L., Wu, M., & Dick, W. A. (2012). Comparison of three methods for detection of melamine in compost and soil. Science of the total environment, 417, 255-262.

Lines 173-175 appear to be an unintentional combination of multiple sentences.

As per the reviewer valuable suggestion the sentence has been edited

What is "it" in line 256?

As per the reviewer valuable suggestion the sentence has been edited

What digitization? What is Industry 4.0? (line 263)

As per the reviewer valuable suggestion the sentence has been edited

Biosensors convert the digital signal intro readable format

Digitization is the process of converting information into a digital (i.e. computer-readable) format. The result is the representation of an object, image, sound, document or signal (usually an analog signal) by generating a series of numbers that describe a discrete set of points or samples.

Figure 4 doesn't seem to show anything coherent. The diagrams are barely legible, the contents of the diagrams are unclear, the purpose of the 3 boxes opaque and the logic of the internal components suspect. Why is there a WBSN interface? What is an WBSN interface? It is never defined. In some cases boxes in the diagram are unconnected (battery, e.g.) and in others the purpose unclear. FSM? GPS? 

As per the reviewer valuable suggestion the figure 2; 3; 4a,b,c; 5; 6 has been edited with detailed expalanation has been incorporated

Figure 2: It shows a diagrammatic illustration of bio-electrode to improve bio-sensor for the hydrogen peroxide grit in optimal condition. (Singh et al. [2] described a disposable biosensor capable of rapidly detecting not only H2O2 but also azide in biological samples utilizing a CAT/PANi/ITO electrode as a bioelectrode. This film is extremely effective at keeping enzyme activity and preventing it from escaping the film. This indicates that this effective film can be used to immobilize not only catalase but also other enzymes and bioactive chemicals, making it a potentially useful platform for the development of biosensors).

Figure 3: Schematic illustration of different types of nanomaterials used in biomedicine

Figure 4a. Application scenario for the proposed iHome Health-IoT system.

Figure 4b. Architectural design of the iMedBox

Figure 4c. Integrated HIS enabled by the proposed iHome Health-IoT system.

This paradigm shift has three aspects, as seen in Figure 4c.

1) Healthcare information systems (HIS) inter-organizational implementation: The information systems (ISs) [87] of all stakeholders participating in the Health-IoT production chain form the backbone of the iHome Health-IoT framework. Cloud computing [87] has provided a viable environment for such inter-organizational con-vergence against the so-called Health-IoT-in-Cloud.

2) HIS cross-border development: The iHome Health-IoT system's in-home termi-nal, iMedBox, serves as a connection between in-home healthcare devices and the HIS. The stakeholders' ISs can be efficiently expanded to a patient's home by installing indi-vidual applications in the iMedBox. As a result, the IHIS exists in the form of both busi-nesses and consumers' homes as the so-called Health-IoT-at-Home. This is in line with the previously stated pattern of medical services shifting from hospital-based to home-based [85].

3) HIS personalization: Personalized programs will be the predominant method of healthcare in the future. Wearable biomedical applications with ultra-low power and low cost, such as Bio-Patch, allow personalized HIS access to patients' bodies, re-sulting in the so-called Health-IoT-on-Body. As a result, HIS knowledge and contact can be handled at the level of a single individual's body.

We created the Health-IoT platform with the current and future relevance of IHIS and IoT in the e-health sector in mind. It can be used in patient's homes and nursing homes. The proposed device combines SoC technology, material technology, and ad-vanced printing technology to create a patient-centric, self-assisted, fully automated in-telligent in-home healthcare solution. Environmental surveillance, vital sign acquisition, drug control, and healthcare facilities are only a few of the cases where the established functions can be used. The creation of several smart devices, including Bio-Patch, iMedPack, and iMedBox, to realize the vision of iHome Health-IoT is de-scribed in this article.

Figure 5. Unobtrusive biosensor impact classification and health monitoring system (Point-of-care(POC) diagnostics play an important role in delivering health care,particularly for clinical management in different organ and surveillance) Brief sensor platform for health monitoring. A major distinction can be made between non-invasive and invasive applications, including wearable sensors for monitoring biophysical, biochemical, environment signals, and implantable devices for nervous, cardiovascular, digestive, locomotors system.

Figure 6. Overview on the working of flexible devices for P.O.C. diagnosis. (We discuss recent advancements in the multiplexing capabilities and sensitivity of paper-based POC diagnostics, and then discuss the development of flexible polymer-based POC diagnostics for sensing biological targets, as well as their prospective uses as wearable devices)

Point-of-care (POC): patient care in the emergency room, in primary clinics, at home, or in other nonhospital settings where diagnosis can be made and treatment can be administered. Surface-enhanced Raman scattering (SERS): a spectroscopic approach that can detect chemical and biological species down to a single molecule without labeling.

In the paper, acronyms are unevenly used and often not defined. Electrocardiogram (ECG) is sometimes E.C.G. The correct way is to omit the periods, as in ECG. Correct all acronyms.

As per the reviewer valuable suggestion the abbreviation has been edited and corrected uniformly

The discussion of the iMedbox structure (line 303 onward) is nonsense and references patents for charger plugs, describes features not shown on Fig 4 and so on.

As per the reviewer valuable suggestion the Figure 4 has been edited and segregated into different part

The so called iTag (line 330) has no breathtaking characteristics as described. It is not described at all.

As per the reviewer valuable suggestion the abbreviations has been edited

[The Interactive Technologies (iTAG)iTag comprises a of Wireless Body Sensor Networks (WBSN)]

  1. Sodhro, A. H., Pirbhulal, S., Qaraqe, M., Lohano, S., Sodhro, G. H., Junejo, N. U. R., & Luo, Z. (2018). Power control algorithms for media transmission in remote healthcare systems. IEEE Access, 6, 42384-42393

Movement antiques? (line 375)

As per the reviewer valuable suggestion the sentence has been edited

I'm halting reading of this.

Kindly Excuse of the inconvenience, as per the reviewer valuable suggestion entire manuscript has been edited into readable format

Reviewer 3 Report

Development of Nano-sensors based Intelligent packaging systems: Food quality and medicine

The abstract does not really provide an indication of the material covered in this review; it is difficult to follow and list a number of topics without clearly explaining the theme; in addition the English makes it difficult to follow e.g. line 28 “Suh as a wireless link, wearable biomedical sensors may collect various vital parameters”  I do not understand what this is trying to tell me partly because the English is confused and partly because “various parameters” is rather vague.

In a similar way the introduction makes a lot of grand statements which are not well supported e.g. line 45 onward “In numerous areas such as healthcare, grafts, prostheses, smart fabrics, power generation and maintenance with power-producing and extremely well-organized arrays, such as, defense, protection, sabotage, and shadowing, nanomaterial applications have ended their presence intensely sensed [1]”  All these surely merit more than one reference; and the last phrase is poorly worded. The introduction generally fails to set the scene for the remaining discussion. 

In the main text the discussion is generally too unfocused it is not clear if this is a review on food packaging or medicine or biosensors or all of these.  This needs to be restructured with more focus and more detail of the topics being discussed.  In addition, the English needs improving.  As a minimum a major rewrite will be needed.

Author Response

Respond to Reviewer-3 Comments

Development of Nano-sensors based Intelligent packaging systems: Food quality and medicine

 The abstract does not really provide an indication of the material covered in this review; it is difficult to follow and list a number of topics without clearly explaining the theme; in addition the English makes it difficult to follow e.g. line 28 “Suh as a wireless link, wearable biomedical sensors may collect various vital parameters”  I do not understand what this is trying to tell me partly because the English is confused and partly because “various parameters” is rather vague.

The authors are really grateful for the reviewer valuable suggestion, towards formatting the manuscript in a readable format for the readers

The abstract has been edited into simple description format 

The issue of medication noncompliance has resulted in major risks to public safety and financial loss. The new omnipresent medicine enabled by the Internet of Things offers fascinating new possibilities. Additionally, an in-home healthcare station (IHHS) is necessary to meet the rapidly increasing need for routine nursing and on-site diagnosis and prognosis. This article proposes a universal and preventive strategy to drug management based on intelligent and interactive packaging (I2Pack) and IMedBox. The Controlled Delamination Material (CDM.) seals and regulates wireless technologies in novel medicine packaging. Suh, wearable biomedical sensors may capture a variety of crucial parameters via a wireless communication. On-site treatment and prediction of these critical factors are made possible by high-performance architecture. The user interface is also highlighted to make surgery easier for the elderly, disabled, and patients. Land testing incorporate and validate an approach for prototyping I2Pack and iMedBox. Additionally, sustainability, increased product safety, and quality standards are crucial throughout the life sciences. To achieve these standards, intelligent packaging is also used in the food and pharmaceutical industries. These technologies will continuously monitor the quality of a product and communicate with the user. Data carriers, indications, and sensors are the three most important groups. They are not widely used at the moment, although their potential is well understood. Intelligent packaging should be used in these sectors, as well as the functionality of the systems and the values presented in this analysis.

In a similar way the introduction makes a lot of grand statements which are not well supported e.g. line 45 onward “In numerous areas such as healthcare, grafts, prostheses, smart fabrics, power generation and maintenance with power-producing and extremely well-organized arrays, such as, defense, protection, sabotage, and shadowing, nanomaterial applications have ended their presence intensely sensed [1]”  All these surely merit more than one reference; and the last phrase is poorly worded. The introduction generally fails to set the scene for the remaining discussion. 

As per the reviewer valuable suggestion entire manuscript has been edited into readable format

In the main text the discussion is generally too unfocused it is not clear if this is a review on food packaging or medicine or biosensors or all of these.  This needs to be restructured with more focus and more detail of the topics being discussed.  In addition, the English needs improving.  As a minimum a major rewrite will be needed.

As per the reviewer valuable suggestion entire manuscript has been rewritten into readable format

Round 2

Reviewer 1 Report

The authors have carefully considered all comments and the responses provided are satisfactory. I recommend the publication of the new manuscript.

Author Response

The authors were grateful for reviewer valuable le comments, as per the reviewer valuable comments, the entire manuscript was edited for grammatical error and rectified

Reviewer 3 Report

This is now suitable but there are still some minor errors in the English which could be dealt with at the editorial phase.

Author Response

(The authors gave the same response as above.)
